# The Evolution of Sustainable Development Theory: Types, Goals, and Research Prospects

**Longyu Shi [1], Linwei Han [1,2] , Fengmei Yang [1,2] and Lijie Gao [1,\*]**

[1] Key Lab of Urban Environment and Health, Institute of Urban Environment, Chinese Academy of Sciences, Xiamen 361021, China; lyshi@iue.ac.cn (L.S.); lwhan@iue.ac.cn (L.H.); fmyang@iue.ac.cn (F.Y.)
[2] University of Chinese Academy of Sciences, 100049 Beijing, China
\* Correspondence: ljgao@iue.ac.cn; Tel.: +86-592-619-0676

**Abstract:** Sustainable development (SD) has become a fundamental strategy to guide the world's social and economic transformation. However, in the process of practice, there are still misinterpretations in regards to the theory of SD. Such misinterpretations are highlighted in the struggle between strong and weak sustainable development paths, and the confusion of the concept of intra-generational and inter-generational justice. In this paper, the literature survey method, induction method, and normative analysis were adopted to clarify the gradual evolution and improvement process of the concept and objective of SD, to strengthen the comprehensive understanding of the SD theory. Moreover, we also tried to bring in the situation and concepts of China. The results show that the theory of SD has gone through three periods: the embryonic period (before 1972), the molding period (1972–1987), and the developing period (1987–present). SD is gradually implemented into a global action from the initial fuzzy concept, including increasing practical wisdom. The goal of SD evolves from pursuing the single goal of sustainable use of natural resources to Millennium Development Goals (MDGs) and Sustainable Development Goals (SDGs). This paper argues that the theory of strong sustainability should be the accepted concept of SD. Culture, good governance, and life support systems are important factors in promoting SD.

**Keywords:** sustainable development; sustainable development goals; culture; governance

## 1. Introduction

In the ancient agricultural economy period, in order to coordinate agricultural development and human survival, the ancient thought of simple and sustainable development (SD) began to sprout [1]. Since the start of the Industrial Revolution, the population has increased rapidly and production has been developing. Human beings have been exploiting wealth from nature and the volume of wastes and pollutants thrown into the environment has also gradually increased. Preserving the global life support systems has become more difficult due to the rapid and continuing human-caused environmental changes [2]. Meanwhile, these changes posed a serious threat to the survival of human beings [3]. The most well-known example is the Eight Major Pollution Incidents in the early 20th century [4]. According to statistics, the Belgian Meuse Valley Fog disaster of 1930 injured thousands of residents and killed more than 60 people within a week [5]; the Donora Smog tragedy in 1948 sickened nearly 6000 people in 5 days [6]; and the Great Smog of London in 1952 killed more than 4000 people in 4 days [7]. At the same time, mankind was faced with worsening problems such as food shortage, energy crisis, and environmental pollution, intensifying "ecological crisis", slowing economic growth, and raising the local social unrest [8]. Such problems have forced humanity to re-examine its position in the ecosystem and search for a new path for long-term survival and development [9]. In this context,

the concept of SD emerged and became a fundamental strategy to guide the world's socio-economic transformation [10].

However, the lack of understanding of SD is still a problem that people need to face whether in academia, government agencies, or private enterprises [11–13]. Currently, most of the definitions and interpretations of SD do not come from the comprehensive concepts of SD. Instead, they are influenced by the basic principles of specific groups or organizations of SD [14–16]. For instance, the description of development does not distinguish between goals and means, but interprets SD as a simple process of change that can last infinitely. Examples of such interpretation could be considering the high growth rate of agricultural production in South Africa as SD [17], or considering SD as an enterprise's business affairs, with a serious tendency to commercialization and pursue theories in the process of sustainable city construction [18,19]. Therefore, it is necessary to revamp the definition of SD [20].

According to the principle of relational dialectics between practice and cognition, and the principle of repetition and infinity of the cognition process, SD practice is the foundation of SD theory [21,22]. SD theory guides SD practice, and is constantly tested and perfected in practice [21]. In order to better guide the SD practice, it is necessary to reorganize the evolution process of the practice and theory of SD. Based on the above analysis, this paper adopted the literature survey methods by reading literature pieces and materials related to SD practice and theory, and the induction method by analyzing related thoughts. Normative analysis refers to the process of making recommendations about what action should be taken or which viewpoint should be taken on a topic, and the positive analysis uses scientific principles to determine the objectives and testable conclusions [23]. Therefore, the normative analysis method was adopted to draw a conclusion based on the existing thoughts on SD. Finally, based on the review of the evaluation of SD theory and practice including SD types and goals, several key issues in the current theoretical research were selected for discussion and prospect.

## 2. Status of SD Research

First, we analyzed the status of SD research to overview the research trend. We chose the ISI Web of Science (WOS) as the data source for its wide application in congeneric articles [24,25]. In order to guarantee the quality of literature, the Core Collection database was selected. In the search process, as SD and sustainability were not strictly distinguished [25–27], we used both "sustainability" and "sustainable development" expressions to avoid omitting relevant literature. To focus on the content related to theory or practice, we used "NEAR/3" to control the distance between SD and theory or practice that is no more than three words. To track the recent trend of SD theory and practice, the retrieval time span was set from 2000 to 2019. After screening the title, abstract and keywords on the relativeness toward SD theory or practice, a total of 1942 papers were retrieved. The details about paper numbers are shown in Figure 1.

In this section, Citespace software was used to set each year as a data segmentation era to conduct a visual analysis of keywords and research hotspots in the literature. The result showed that the country with the most articles on SD theory or practice were published in the United States, following by China (Figure 2). Research hotspots are mainly focused on SD management in different fields, including small and medium enterprises management, environmental protection, sustainable tourism, and sustainability assessment. China is a hot research area (Figure 3). Although studies on SD theory or practice have been increasing from 2000 to 2019, they were mainly focused on the aspect of the practice. The studies on SD theory are fewer than SD practice, and the trend of articles published tends to be stable or even decline (Figure 1). However, the study of SD theory has important guiding significance for SD practice, therefore, it is necessary to further strengthen the study of SD theory.

## 3. The Evolution Process of SD Theory

SD theory developed through practice, and the study of SD cannot be separated from the implementation of relevant policies [22,28]. SD has experienced the germination of ideas, and then a series of SD practices, such as the United Nations Sustainable Development Summit. Many changes

have taken place and SD has evolved from tackling environmental issues to deal with the global strategic issue [29]. Based on the study of the evolution of SD thoughts and the formation of SD theory by Lele [17], Mebratu [14], Zhang [30], and other scholars, this paper divided the evolution and development of SD theory into the embryonic period (before 1972), the molding period (1972–1987), and the developing period (since 1987) (Figure 4).

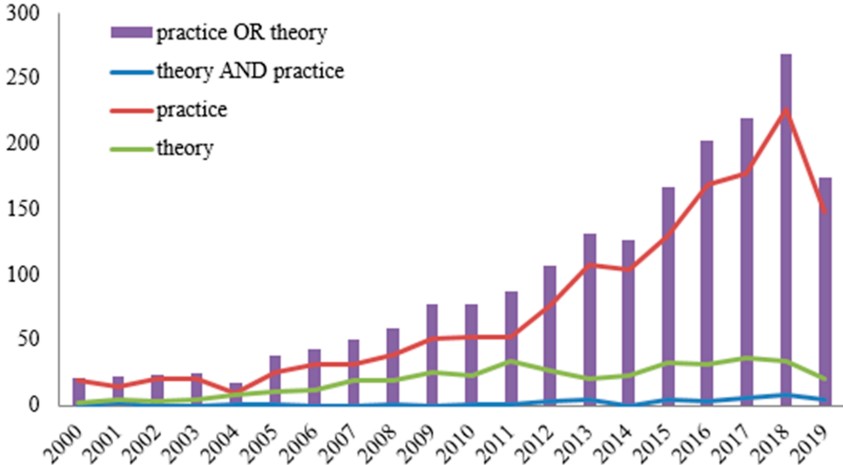

**Figure 1.** Number of papers on "sustainable development" published from 2000 to 2019 in Web of Science (WOS) core database (**Note:** The retrieval formula of "theory" is: TS—(("sustainable development" OR sustainability) NEAR/3 theory) NOT TS—(("sustainable development" OR sustainability) NEAR/3 practice); The retrieval formula of "practice" is: TS—(("sustainable development" OR sustainability) NEAR/3 practice) NOT TS—(("sustainable development" OR sustainability) NEAR/3 theory); The retrieval formula of "practice OR theory" is: TS—(("sustainable development" OR sustainability) NEAR/3 theory) OR TS—(("sustainable development" OR sustainability) NEAR/3 practice). The retrieval formula of "practice AND theory" is: TS—(("sustainable development" OR sustainability) NEAR/3 THEORY) AND TS—(("sustainable development" OR sustainability) NEAR/3 practice)).

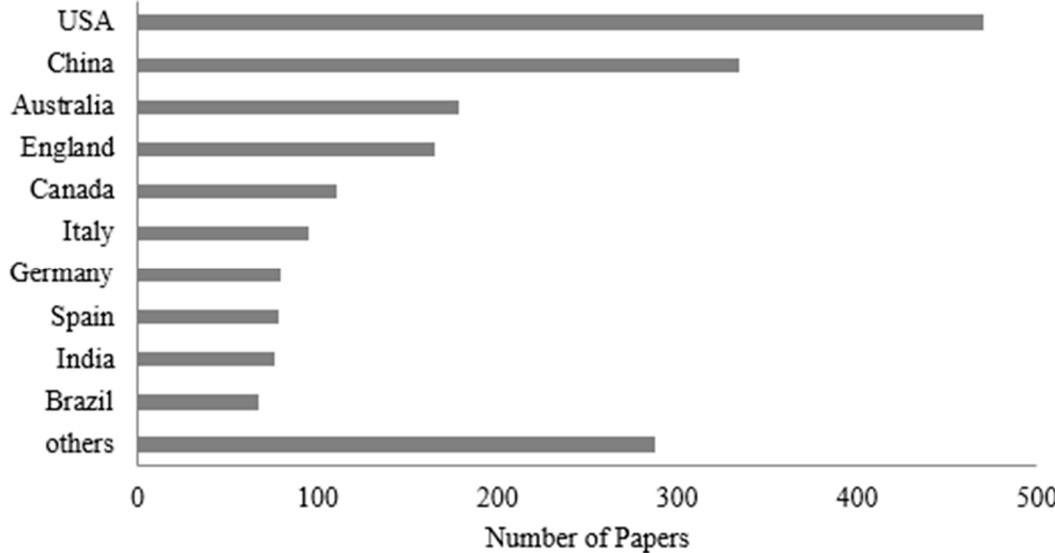

**Figure 2.** Main research countries on sustainable development (SD) theory and practice.

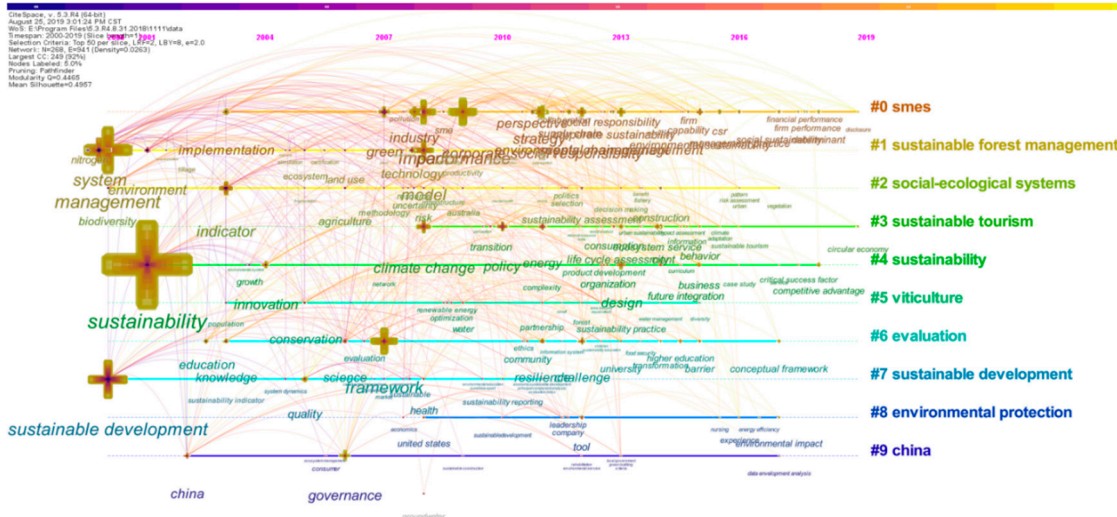

**Figure 3.** Clustering analysis of keywords of research of "practice AND theory" from 2000 to 2019 (**Note:** "smes" is small and medium enterprises).

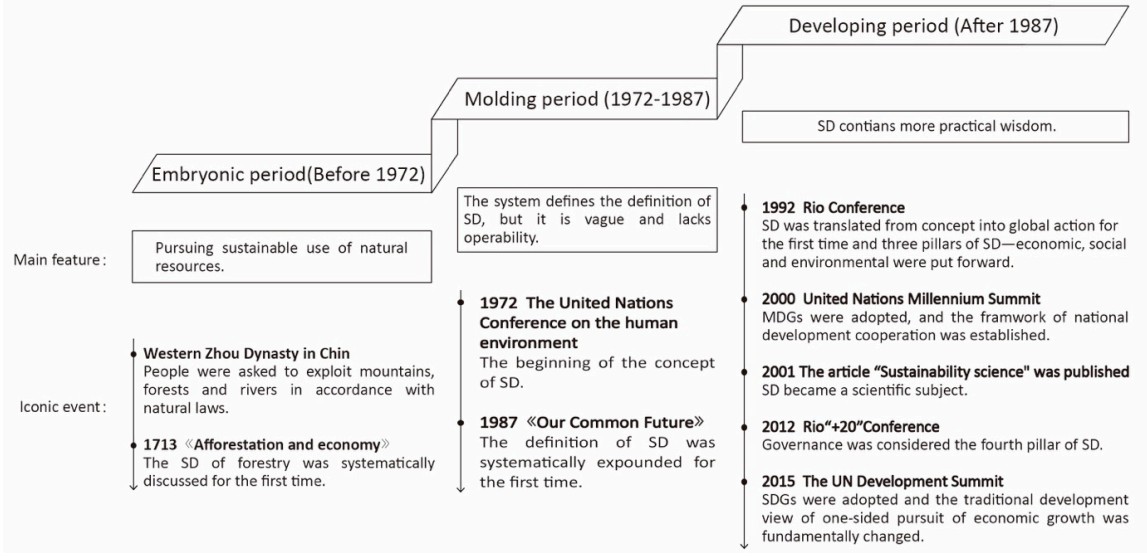

**Figure 4.** Evolution stages and symbolic events of SD theory.

### 3.1. The Embryonic Period (Before 1972)

Though the terms "sustainability" or "sustainable" first appeared in the 20th century, the equivalent concepts have been used for centuries. The idea of SD has a long history in China, and the concept of SD can be traced back to the core idea of Chinese classical philosophy—"天人合一" (Heaven and people in one) [31]. As early as the Western Zhou Dynasty (B.C.1100~771), the emperors realized that the mountains, forests, and rivers should be rationally used according to the laws of nature rather than overexploiting it. Since then, various measures have been taken to protect natural resources, such as establishing designated management departments, enforcing state monopoly, collecting taxes in regard to mountains and gardens, and issuing bans [1]. Afterward, different causes of environmental degradation, including farming, logging, and mining, were discussed in the ancient Egyptian, Mesopotamian, Greek, and Roman civilizations [3]. And some measures were also recommended. Varro (first century AD) stated that "we can, by care, lessen the evil effects" [32]. The term "sustainability" was first used in Carlowitz's monograph — *Afforestation and Economy*, which was published in 1713 and systematically addressed the issue of sustainable forestry [33].

The *German Dictionary,* which was published in 1809, interpreted the term "sustainability" as something that people can still rely on when everything else is unsustainable. It can be seen that the early thought of SD mainly reflected in the sustainable use of natural resources.

*3.2. The Molding Period (1972–1987)*

3.2.1. The United Nations Conference on the Human Environment in 1972

With the recognition that the developmental policies primarily focusing on economic growth increased the frequency of serious environmental problems, the United Nations held a world summit in Stockholm, Sweden in 1972. The conference was the first human environment conference and symbolized the beginning of the SD concept [34]. The summit urged all countries in the world to strengthen environmental management policies while developing their economies. Since then, institutional developments related to environmental protection and vigorous end-of-pipe environmental governance have advanced. However, there was a major global disagreement at the summit, i.e., the northern developed countries were more concerned with environmental issues, while the developing countries in the south paid more attention to poverty issues.

3.2.2. Publication of "Our Common Future" in 1987

In 1987, the World Commission on Environment and Development (WCED) drafted a report on human development, "Our Common Future", which was the first time to systematically stated the definition of SD. SD was defined as "sustainable development is development that meets the needs of the present without compromising the ability of future generations to meet their needs" [35]. The report focused on the global situation of the population, food, species and genetic resources, energy, industry, human habitation, etc. In addition, the report systematically discussed a series of major economic, social, and environmental issues faced by humanity and clearly proposed three viewpoints: 1) the crisis of environmental, energy, and development cannot be separated; 2) the resources and energy on earth are insufficient for the needs of human development; and 3) current developmental models must be changed for the interests of present and future generations [35]. These definitions and viewpoints are highly general and concise but lack direct and practical operability.

*3.3. The Developing Period (1987–Present)*

3.3.1. The 1992 United Nations Conference on Environment and Development

In 1992, the United Nations hosted a conference, in Rio de Janeiro, Brazil, to address environment and development, starting the journey of SD in a global scope. The conference passed and signed the "Rio Declaration on Environment and Development" and the "Agenda 21". Moreover, one agreement was identifying the "common but differentiated responsibilities" of developed and developing countries in addressing global environmental issues, as well as the need for developed countries to finance and transfer technology to developing countries. This meeting has also formulated goals and action plans to implement sustainable development and establish the principle of building a global partnership to jointly solve global environmental problems [34]. This was the first time in human history that SD strategy has been implemented from a concept into a global action [36], which established the importance of SD at the international policy level. Since then SD has become the consensus of the whole of mankind [37]. Furthermore, the conference treated SD as a core concept for resolving the apparent contradiction between economic development and environmental protection, pointing out that SD involves development in a sustainable manner regarding resources and the environment. The conference also emphasized the societal polarization and the importance of equity, thus introducing the social dimension of the theory of SD. Based on this theoretical advance, the three pillars of SD were considered to be economy, society, and environment [38].

### 3.3.2. The United Nations Millennium Summit in 2000

In September 2000, the United Nations Millennium Summit was held at the United Nations Headquarters in New York. The representatives of 189 countries adopted the "United Nations Millennium Declaration", which identified the Millennium Development Goals (MDGs) with the development and elimination of extreme poverty as the focus, including eight key areas and 21 operational targets [39]. These goals became an internationally recognized framework for guiding national development and cooperation over the next 15 years and provided guidance for the development of humanity in the new century.

### 3.3.3. From SD to Sustainability Science

Despite the great political achievements toward SD, many scientists found it difficult to conceptualize and measure SD [29,40]. In 1999, the National Research Council (NRC) published a report named "Our common journey: A transition toward sustainability". The report put forward the word "sustainability science" and explained it as "the science of sustainable development" [41]. In 2001, the article "Sustainability Science "could be seen as a milestone since the birth of Sustainability Science [42]. The paper pointed out that sustainability science was aimed to explain the interaction between natural and social characteristics and to improve the ability to steer this interaction toward a more sustainable trajectory. Since then, sustainable development has become a scientific subject covering agriculture, ecological economics, forestry, etc. [12,43].

### 3.3.4. The 2012 United Nations Conference on SD

After 1992, the conflicts of interest among the economy, society, and environment became increasingly apparent. The need to introduce the concept of cooperative governance of global stakeholders became more crucial [44]. Under this background, the United Nations held the "Rio+20" Summit in 2012. The summit indicated that the green economy was the key to solve conflicts between development and the environment [45]. Moreover, global cooperative governance can solve conflicts among economic, social, and environmental issues. By the end of the summit, SD expanded from three pillars to four: economic, social, environmental, and governance [46].

### 3.3.5. United Nations Sustainable Development Summit

In September 2015, more than 150 heads of state and government participated in the United Nations Sustainable Development Summit at the United Nations headquarters in New York. The summit assessed the implementation of MDGs and adopted "Transforming our World — the 2030 Agenda for Sustainable Development" [47]. The agenda set out the Sustainable Development Goals (SDGs), covering 17 focus areas and 169 specific targets. Compared to MDGs, SDGs changed the traditional concept of development fundamentally. Besides solely pursuing economic growth, SDGs put forward the concept of inclusive growth and SD featuring coordinated economic, social, and environmental development.

## 4. Types of SD

At present, most scholars agree with the classification of natural capital, manufactured capital, human capital, and social capital [27,48]. To achieve SD in human society, it must depend on the stock of four kinds of capital and their relationships within a certain period. The understanding of the relationships among the four types of capital has an important impact on how to interpret and evaluate sustainability [31]. Currently, there are three main interpretations of the mutual substitution between natural capital and manufactured capital (Figure 5).

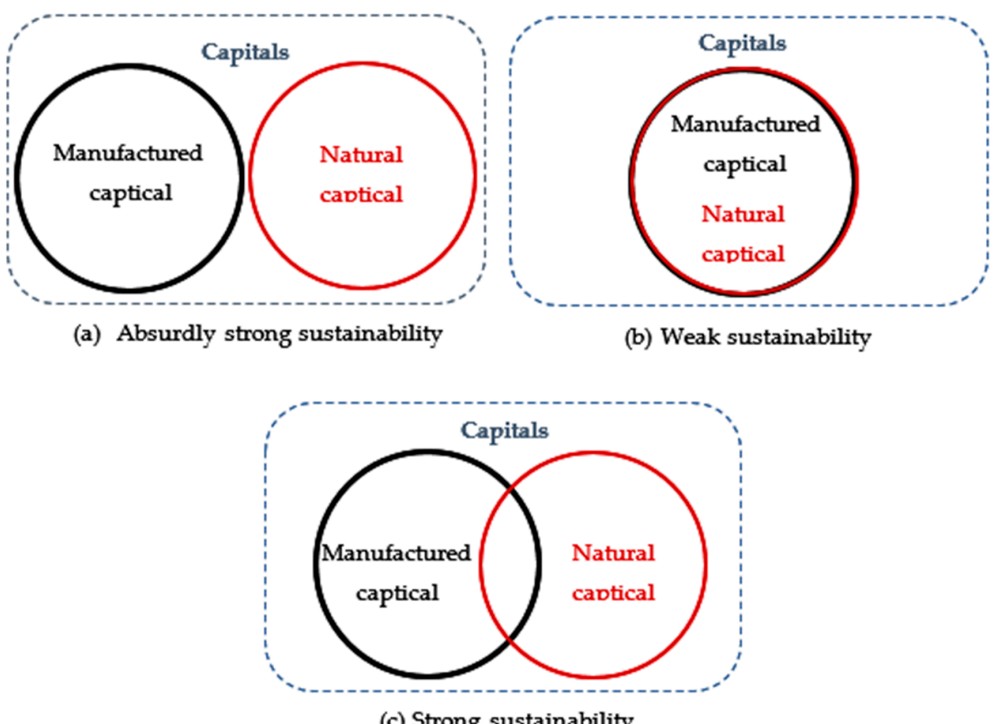

**Figure 5.** Schematic diagram of different types of sustainability (**Note:** the overlapping parts of the two circles represent capital that can be replaced by each other).

### 4.1. Weak Sustainability

Weak sustainability is a human-centered view that natural capital can be replaced by manufactured capital. As an extension of neoclassical welfare economics [27], weak sustainability considers the total amount of manufactured capital and natural capital as the most important [49]. Therefore, as long as the total amount of capital increases in the process of development, even if the natural capital degrades to an unrecoverable state, it is still sustainable [50].

### 4.2. Strong Sustainability

Strong sustainability is a nature-centered view that natural capital plays an irreplaceable role in production and consumption. This concept is mainly based on the steady-state economic theory that manufactured capital cannot be duplicated without the input of natural capital [51]. Therefore, the process of development should not only require an increase in the total amount of capital, but also require the rationality of the capital structure and not crossing ecological thresholds [52]. Moreover, economic development should not exceed the natural limit [53].

### 4.3. Absurdly Strong Sustainability

Absurdly strong sustainability not only believes that natural capital cannot be replaced by manufactured capital, but also believes that the exploitation and utilization of ecosystems should be eliminated [53]. After SD thought was developed, some radical environmentalists put forward the view that human beings and other species are equal [54]. They believe that human beings should develop without changing the status quo of nature. Even more extremely, those radical environmentalists advocate a non-development of human society in exchange for the restoration and maintenance of natural, but such concept is unrealistic [31].

### 4.4. Comparisons Among Weak, Strong, and Absurdly Strong Sustainability

The advantage of weak sustainability is that it affirms the progress of science and technology, the substitutability of natural and manufactured capital is more in line with the needs of economic development. The disadvantage of weak sustainability is that it is too optimistic about human's ability to control nature and technological progress, and it believes that nature has no constraint capacity and that all the ecosystem functions can be replaced. The advantage of the absurdly strong sustainability is that the economic system is a subsystem of nature, rather than an independent one. The disadvantage of the absurdly strong sustainability is that the role of technology is underestimated, and that all the natural capital is thought to be restrictive, while in fact some natural capital can be replaced. For example, some mineral resources can be replaced by other man-made products. The advantage of strong sustainability is that it is opposed the above two extreme views, thus, strong sustainability is the concept that we should accept [55]. The disadvantage of strong sustainability is that it sets a series of thresholds that must not be crossed, which may hinder the economic development of especially developing countries.

At present, there are relatively few in-depth studies on the strong and weak levels of SD in various evaluation indicators and measurement methods of SD. Most of them are based on the view of weak sustainability, that is, the simple sum of three systems (economic, social, and environmental) to measure the SD level. Green GDP, which is calculated by subtracting the costs of environmental and ecological damage from a country's conventional GDP, also has such characteristics.

## 5. Goals of SD

Human society has different development problems in different periods. Under the guidance of the theory of SD, SD practices continue to evolve, and SD goals continue to be enriched and improved [56,57]. The objectives of SD have evolved from the single-factor goal of sustainable use of animal and plant resources to MDGs focusing on poverty, and now to the more comprehensive and universal SDGs.

### 5.1. Single-Factor Development Goals

In the period of the agricultural economy, people fully realized that the regeneration ability of natural resources is limited. Ancient sages advocated restricting the utilization of the resources that human beings rely on for survival, such as forestry, fishery, birds, and animals [1]. 《史记·殷本纪》 (*Records of the Historian—the Yin Dynasty*) embodies the sustainable use of animal and poultry resources. 《论语·述而》 (*The Analects of Confucius*) and 《吕氏春秋》 (*The Annals of Lu Buwei*) reflect the sustainable use of fishery resources, and 《秦律·田律》 (*Qin Law·Tian Law*) reflects the sustainable use of forest resources. In 1713, Carlowitz published his book *Afforestation and Economy* that systematically discussed the SD of forestry [33]. The term sustainable development came into prominence in 1980, when the International Union for the Conservation of Nature and Natural Resources (IUCN) presented the World Conservation Strategy (WCS) with "the overall aim of achieving sustainable development through the conservation of living resources" [58]. Then a series of influential actions, conferences, documents, and discussions, such as the *21ˢᵗ-century agenda*, paved the way for a global mindset on sustainable development [59]. Environmental issues were also recognized as the priority of supports and believed as the foundation and core of sustainable growth by international groups including International Monetary Fund (IMF), World Bank [60,61]. Some SD assessment indicators were also established, including System of Integrated Environmental and Economic Accounting (SEEA), Ecological Footprint (EF). Millennium Ecosystem Assessment (MA) (2001–2005) also assessed the consequences of ecosystem change for human well-being [62]. In short, SD in this period mainly addressed the issue of ecological sustainability with the goal of sustainable utilization of natural resources and environmental protection [17].

### 5.2. Millennium Development Goals (MDGs)

MDGs were the international development goals that had been established by the UN summit in 2000, such as eradicating extreme poverty and hunger, achieving universal primary education, promoting gender equality, and empowering women. Thus, the core of MDGs was poverty alleviation of developing countries, making unremitting efforts to improve the lives, save the lives, and seek the survival of extremely poor developing countries through a set number of years [63].

By 2015, the 15-year time limit for MDGs had expired, and unprecedented results have been achieved worldwide, making MDGs the most successful global anti-poverty program in history. In developing areas, the number of people living in extreme poverty dropped from 1.9 billion to 836 million; the proportion of people living in extreme poverty dropped from 47% in 1990 to 14% in 2015; the net enrollment rate of primary schools in developing regions increased from 83% in 2000 to 91% in 2015; the global under-five mortality rate dropped from 9% in 1990 to 4.3% in 2015; the proportion of the global population has access to improved drinking water source increased from 76% to 91% [64].

Even so, there was still room for improvement between the progress of MDGs and the preset goals. This significant gap mainly reflected in the following aspects: 1) Conflicts remained the biggest threats to human development. 2) The achievement of each subproject's objectives was unbalanced. Although the goal of universal primary education was achieved and halving the proportion of people without access to improved drinking water was accomplished five years ahead of schedule there are still many goals that are not meet. In fact, a third of developing countries did not reach gender equality in their primary education system and the goals of halving the proportion of people affected by hunger, reducing child mortality by two-thirds, reducing maternal mortality by two-thirds, and halting and beginning to reverse the spread of HIV/Aids were not met. 3) MDGs progress in each region was uneven. For instance, East Asia and South America fared much better than Africa. Despite noticeable progress, there were still large gaps between rural and urban, and between the poorest and richest households. 4) There was a large gap between the efforts and performance of developed countries and their commitments. Developed countries had made several international public pledges of assistance to developing countries, but most of these pledges had not been delivered on time. 5) The focus of MDGs was to solve the survival problems faced by the extreme poverty in developing countries, but due attention was not paid to how to guarantee SD [65–67].

### 5.3. Sustainable Development Goals (SDGs)

Based on the experience of MDGs, the UN adopted SDGs in 2015 [68]. SDGs incorporate six elements: dignity, human beings, the planet, prosperity, justice, and partnership. SDGs are comprised of 17 goals and 169 sub-goals to guide the SD for all regions, including developed and developing countries, in the next 15 years [69,70].

SDGs can be divided into four aspects: economy (goals 8, 9, 10, and 12), society (goals 1, 3, 4, 5, 11, and 16), environment (goals 2, 6, 7, 13, 14, and 15), and governance (goal 17) [71]. SDGs have been compared with MDGs by scholars, and the main differences are as follows: 1) SDGs are more universal than MDGs. MDGs were mainly targeted at developing countries, while SDGs cover countries with high, middle, and low income [72]. 2) SDGs are more comprehensive and specific than MDGs. For example, both of them attached great importance to global environmental security, but MDGs generally put forward "ensuring environmental sustainability", while SDGs put forward specific goals such as combating climate change, conserving and sustainably using the oceans, seas, and marine resources, etc. [73]. 3) SDGs have higher standards than MDGs. For example, the goal of MDGs for poverty is to "eradicate extreme poverty and hunger", while the goal of SDGs is to "end poverty in all its forms everywhere". Moreover, while MDGs focused on a higher enrollment rate of school, SDGs propose to improve the quality of education. 4) SDGs pay more attention to the bidirectional nature of cooperation than MDGs did. MDGs emphasized sending aid from developed countries to developing countries, while SDGs emphasize subjective initiative and responsibility of all

parties. 5) Data revolution. Goal 17 of SDGs proposes to enhance the ability to obtain high-quality, timely, and reliable data. This concept was not mentioned in MDGs. 6) Transformation of the development paradigm. Compared with the MDGs, SDGs call on people who live in extreme poverty in developing countries to not only survive, but also live with dignity. The core purpose of SDGs is that in the process of achieving poverty alleviation and promoting economic development, people should not survive at the cost of damaging the ecological environment, and must adhere to and implement the concept of SD [74].

According to *The Sustainable Development Goals Report 2018* released by the United Nations Department of Economic and Social Affairs, over the past three years, the pace of progress in various development fields has been slow and uneven, making SDGs goals difficult to meet by 2030. Moreover, the lack of funding is one of the main obstacles to achieve SDGs. The report shows that in 2017, the net official development assistance was 146.6 billion dollars, a 0.6% decrease compared with that in 2016. Official development assistance as a percentage of donor countries' Gross National Income (GNI) remained low at 0.31%. Thus, the international community needs to create conditions to mobilize countries to undertake internal tax reform and, on the other hand, the international community needs to combat tax evasion and the illegal flow of capital.

## 6. Prospects of SD Research

At present, although the theory of SD has been widely promoted, in practice, many theoretical and methodological problems of SD have not been fundamentally solved. According to the previous analysis, although strong sustainability is the concept people should accept, the practice of weak sustainability remains. Even though the goals of SD have developed from addressing ecological sustainability to more comprehensive goals, those goals only consider SD within the next 15 years rather than a longer term. Furthermore, in 1992, the three pillars of SD were the economy, society, and environment. Some authors have introduced the fouth pillar of culture, institutions, or governance [75–77]. Based on the above analysis, this paper selected four issues for further discussion.

### 6.1. Weak Sustainability Remains

Although strong sustainability is the concept of SD we should accept, weak sustainability is still common. This phenomenon is due to factors such as cognition, different understanding levels of SD and levels of social and economic development. While criticizing the malpractices of human society, the weak sustainability model acknowledges the partial rationality of the existing world and is willing to compromise. This is a development model that is more psychologically acceptable to contemporary people [78]. For example, China's green GDP is an index that determines GDP after deducting the loss caused by environmental damage. If the result of the green GDP is low or negative, it means that economic is developed at the cost of environmental damage, which will not be sustainable, otherwise, economic development will be considered to be sustainable. However, this reasoning presumes that there is a complete substitution between natural resources and other resources, which is an idea that is supported by weak sustainability. Under this thinking, countries tend to artificially lower the price of natural resources to promote economic growth [79]. Similarly, in 1995, the World Bank proposed the term "Genuine Saving" (GS) to predict the SD of a country's economy. This term took into account the impact of depletion of natural resources, damage to environmental pollution and investment of human capital on national wealth [80]. However, it was still examining the total amount of capital. Although negative GS is equivalent to unsustainable, positive GS does not necessarily equal to sustainable [81].

The purpose of SD is to maintain the increase of the total capital amount and the rationality of the capital structure during the development process. It is impossible to totally decouple GDP growth from environmental impact [82–84], that is, nonconsumption of natural capital is unacceptable to all countries. But according to strong SD, people should develop within planetary boundaries [84]. Therefore, in the short term, we should practice strict economy, eliminate waste, and promote the construction of a conservation-oriented society. In the long run, we should put efforts to improve the

human capital to replace natural capital, including replacing fossil energy with renewable energy such as solar or wind energy [85]. And for developing countries, additional policies addressing primary production, rural poverty, etc. are needed [86].

### 6.2. Inter-Generational Equity in SD

Equity is an important issue in policy and the field of SD research. Equity of SD is mainly reflected in four aspects: first, intra-generational equity, that is, equity among different social groups in the use of resources and the distribution of products. Second, inter-generational equity, that is, the equity of the right to development between the present and the next generation. Third, procedural fairness, that is, the fairness of various decision-making procedures and political rules. Fourth, species equity, that is, the equity of the right to survival and the right to reproduction between human beings and other species [19,87]. The current interpretation of equity in SD mainly focuses on intra-generational equity. For example, both MDGs and SDGs only consider the application of SD within the next 15 years. From the perspective of the principle of fairness, the distribution of benefits and pressures under such short-term goals may affect the establishment of long-term goals, and may not guarantee the welfare of future generations [88]. Therefore, the need to pursuit intergenerational equity and establish a more adequate view of intergenerational equity is crucial to the realization of the SD goals.

### 6.3. The Cultural Dimension of SD

The preface to *Universal Declaration on Culture Diversity* defines culture as: "the set of distinctive spiritual material, intellectual and emotional features of society or a social group, and that it encompasses in addition to art and literature, lifestyles, ways of living together, value systems, traditions and beliefs" [89]. At the Rio+20 Summit, although culture was not the main theme, its importance for development was recognized [90]. In the process of SD, one standard often fails to solve multiple problems [91]. Many well-intentioned development projects are hampered by a lack of consideration of local realities, cultural identity and values [92]. At present, the cultural dimension of SD has been paid more and more attention. On 17th November, 2010, the University of California, Los Angeles (UCLA) executive board approved the policy statement "Culture: Fourth Pillar of Sustainable Development"; it was within the framework of the third world congress of the world summit of local and regional leaders in Mexico City [77]. However, how to use culture to promote SD still needs further operational guidance.

### 6.4. Cooperative Governance for SD

Governance can be understood as a model of social coordination. Different from ruling behaviors (purposeful guidance, control, and management), governance is how one acts through multiple types of interactions (deliberation, negotiation, self-regulation, or authority selection) and to what extent the participants adhere to collective decision-making [93]. Basic services of governance include the protection of property rights and the smooth operation of the legal system. On the other hand, efficiency, effectiveness, rule of law, participation, accountability, transparency, respect for human rights, and tolerance of differences are also part of governance [94]. Many scholars believe that good governance is a prerequisite and the main factor for achieving SD [95–97]. However, no single form of governance can achieve sustainability in practice [93,96]. Governance varies with different environment and culture and develops through the development of social culture and economy [31,98].

Improving governance capacity and levels requires action in multiple areas, and not all areas can be addressed simultaneously or achieve global consensus. There is a new consensus to accept differences in the development and improvement of governance systems, but it is still necessary to reconfirm the global norms and member states' standards [64]. United Nations Committee of Experts on Public Administration (UN CEPA) and United Nations Department of Environmental and Social Affairs (UN DESA) considered the different governance structures and the conditions of different countries, the development abilities and levels based on respecting national policies and priorities. As a

result, 11 basic principles of governance were formulated: effectiveness (ability, good decision-making, and cooperation), accountability (integrity, transparency, and independent oversight), and inclusiveness (no one left behind, no discrimination, participatory, supportive, and intergenerational equity). In addition, clear and measurable goals and indicators are particularly important for governance with complex concepts and connotations.

### 6.5. Global Life-Supporting System

Though the focus on Earth's life-supporting system is not a comprehensive alternative to other existing efforts to improve life quality, the stable functioning of the global life-supporting system is a prerequisite for human development [2,99]. However, since 2000, accumulating research has shown that this functioning is at risk and that human pressure may lead to further changes in climate, biodiversity, land use, etc. [100,101]. Some research shows that more than 50% of Earth's land surface has been directly modified by human action by 2012 [102]. MDGs concentrated largely on social outcomes; unlike MDGs, SDGs explicitly incorporate objectives to preserve natural ecosystems [99]. To achieve this goal, it is important to find ways, such as the Chinese Yin-Yang approach, to balance the Anthropocene and Ecocentric aspects of SD [103].

## 7. Conclusions

The theory of SD appeared in the 1980s, focusing on the coordinated development of economy, society, and environment, and has entered the high-level political agenda. Currently, SD theory has become an integral part of the agenda of governments and companies. SD goals have become a core part of research institutions' missions around the world [26]. The theory of SD has experienced different stages of development since it was put forward. At present, there are various definitions of SD, but misinterpretations still exist. Based on the processes of cognitive development, this paper combines the evolution of practice and theory, and concludes that the theory of SD has gone through three stages: the embryonic stage (before 1972), the molding stage (1972–1987), and the developing stage (1987–present). The concept of SD has gradually evolved from the initial vague definition to a global action and has contained increasing practical wisdom. In the process of development, it is considered that strong SD, which requires the total capital increase and rationality of capital structure, is the concept of SD people should accept. The goal of sustainable development has become more comprehensive and universal, changing from the single factor goals focusing on ecological sustainability to MDGs, and SDGs today. However, at present, there is still a confusion of strong and weak sustainability, as well as inter-generational and intra-generational equity. Weak sustainability and the pursuit of short-term intra-generational equity still exist. In addition, considering local cultural factors, improving governance capacity and focus more on life support systems are considered as important factors to promote sustainable development.

**Author Contributions:** L.S. conceived and designed the methodologies; L.H. collected and analyzed the data; L.S., L.H., and F.Y. wrote the paper; L.G. and L.H. revised the paper.

**Funding:** This work was supported by the National Key Research and Development Program of China, Grant No. 2018YFC0704703, the "Strategic Priority Research Program (A)" of the Chinese Academy of Sciences, Grant No. XDA23030201, and National Natural Science Foundation of China (General Program), Grant No. 71874174.

**Conflicts of Interest:** The authors declare no conflict of interest.

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
