# Peer review of "The Evolution of Sustainable Development Theory: Types, Goals, and Research Prospects"

_sustainability, doi:10.3390/su11247158_

Round 1
Reviewer 1 Report
The submitted manuscript could make a significant contribution to the field of sustainable development. Although the paper structure and contents appeared to be missing some important elements, it sufficiently described previous work and showed good presentation overall. The authors are encouraged to make major revision to the manuscript as follows:
The title may not best describe what has been written: what about “The evolution of sustainable development theory: Types, goals and research prospects”?
Some relevant references appear to be missing: for example, the first three sentences in Introduction; 1st sentence in the 2nd paragraph on page 2; the first three sentences in 3.3.3; the first two sentences in 4.3; and the first six sentences in 6.3.
Whole methods section is missing. The authors should explain: how they did the literature survey (procedure, why only WOS, how to define literatures and materials related to SD practice and theory, screening criteria, why this particular period between 2000 and 2019, how many papers in total to be searched, how many papers in total to be reviewed, etc.); and normative analysis (what it is, how it is different from positive analysis, how it has been done).
3.1. The embryonic period feels like there is too much emphasis on Chinese ancient/classical philosophy. What about other Asian philosophies? African philosophies? Other more western examples? It appears to be biased.
The last paragraph in 3.3.4. United Nations Sustainable Development Summit needs to be deleted.
Headings, 4. Connotation interpretation of SD and 5. Perfecting process of the objectives of SD, may not be the best headings to describe what was written: 4 could be “Types (or levels) of SD”? 5 could be “Goals of SD”?
Section 4 appears to neglect social dimension of sustainability focusing on economy and environment.
The second paragraph in 4.3. Absurdly strong sustainability could become separate sub-section, 4.4. Comparisons among weak, strong and absurdly strong sustainability. The discussion on advantages and disadvantages of weak, strong and absurdly strong sustainability should be more structured such that all dis/advantages of three types of sustainability are equally discussed.
Please define Green GDP on page 7.
In 5.1. Single-factor development goals, precursors of MDGs are missing such as a series of UN-led conferences in the 1990s, and other activities, documents and discussions by OECD, World Bank and IMF.
In 5.2. Millennium Development Goals (MDGs), most sentences appear to use wrong tenses: shouldn’t they be past or past perfect?
In 5.3. Sustainable Development Goals (SDGs), some verb forms appear to be incorrect: please correct all the verbs according to the plural forms of MDGs and SDGs.
Some English expressions are unconventional: professional proofreading is required. For example, in the second paragraph of 5.2., “Armed conflicts, kidnappings, terrorist attacks, et al.” or in the second paragraph of 5.3., “Some comparations have been made”.
Some sentences are overly long: please split these long sentences into two or more sentences. For instance, “Take China’s green GDP index as an example […] otherwise economic development is sustainable” or “Similarly, the “Genuine Saving” (GS) proposed […] but it still examined the total amount of capital” in 6.1.
In 6. Prospects of SD research, please expand how the authors selected the four issues and introduce the issues before 6.1.
The expressions, “real concept of SD” or “real SD” should be reconsidered as the meaning is not clear: why real? Who defines what is real or not? Is it about ideal? Or sufficiently strong? Sustainable development?
Whole discussion and conclusions are missing. So what? What are the overall lessons learned? What do you suggest after your review?
In author contributions, FY’s contribution is missing: if this is not a mistake, s/he should be removed from the author list.
Author Response
Reviewer 1
Comments:
The submitted manuscript could make a significant contribution to the field of sustainable development. Although the paper structure and contents appeared to be missing some important elements, it sufficiently described previous work and showed good presentation overall. The authors are encouraged to make major revision to the manuscript as follows:
The title may not best describe what has been written: what about “The evolution of sustainable development theory: Types, goals and research prospects”?
Response: Thank you very much for taking time to check our article and putting forward good advice. We think the title you suggested is more proper for this paper and have changed the title according to your advice.
Some relevant references appear to be missing: for example, the first three sentences in Introduction; 1st sentence in the 2nd paragraph on page 2; the first three sentences in 3.3.3; the first two sentences in 4.3; and the first six sentences in 6.3.
Response: Thank you so much for your careful checking. We have added the relevant references where you mentioned.
Whole methods section is missing. The authors should explain: how they did the literature survey (procedure, why only WOS, how to define literatures and materials related to SD practice and theory, screening criteria, why this particular period between 2000 and 2019, how many papers in total to be searched, how many papers in total to be reviewed, etc.); and normative analysis (what it is, how it is different from positive analysis, how it has been done).
Response: Thank you very much for your comment. We have complemented the method section in last paragraph in 1. Introduction and the first paragraph in 2. Status of SD research.
In 1. Introduction, we added: “Moreover, normative analysis refers to the process of making recommendations about what action should be taken or taking a particular viewpoint on a topic, while positive analysis uses scientific principles to arrive at objective, testable conclusions [1], so the normative analysis method was adopted to draw a conclusion based on the existing results of SD thought. Finally, based on the review of the evaluation of SD theory and practice including SD types, several key issues in the current theoretical research on SD were selected for discussion and prospect.”
In 2. Status of SD research, we added: “ISI Web of Science was chosen as the data source in consideration of its wide application in congeneric articles [2]. Furtherly, in order to guarantee the quality of literature, the Core Collection database was selected. In the search process, as SD and sustainability were not strictly distinguished [2–4], we used the expression both ‘‘sustainability’’ and ‘‘sustainable development’’ for not omitting relevant literature. In order to focus on the content related to theory or practice, we used “NEAR/3” to control the distance between SD and theory or practice is no more than 3 words. To display the recent trend of SD theory and practice, the retrieval time span was set from 2000 to 2019. After screening by checking the title, abstract and keywords are whether related to SD theory or practice, a total of 1942 papers were retrieved.”
1. The embryonic period feels like there is too much emphasis on Chinese ancient/classical philosophy. What about other Asian philosophies? African philosophies? Other more western examples? It appears to be biased.
Response: Thank you very much for your good advice. We have condensed the content of Chinese ancient/classical philosophy and added examples from the ancient Egyptian, Mesopotamian, Greek and Roman civilizations, as well as in German.
The revised 3.1 The embryonic period (before 1972) is “Though the terms “sustainability” or “sustainable” appeared in 20th century, the equivalent terms have been used for centuries. The idea of SD has a long history in China, and the concept of SD can be traced back to the core idea of Chinese classical philosophy — “天人合一” (Heaven and people in one)[5]. As early as the Western Zhou Dynasty (B.C.1100~B.C.771), The emperors have realized that the mountains, forests and rivers should be rationally used according to the laws of nature and should not be overexploited. Then various measures have been taken to protect natural resources, such as setting up special management agencies, state monopoly, collecting taxes on mountain and garden, and issuing bans [6]. Afterwards, different environmental degradation were discussed in the ancient Egyptian, Mesopotamian, Greek and Roman civilizations, including farming, logging and mining [7]. And some measures were also recommended. Varro (1st century AD) stated that “we can, by care, lessen the evil effects”[8]. The term “sustainability” was first used in Carlowitz’s monograph —Afforestation and Economy, which was published in 1713 and systematically addressed the issue of sustainable forestry [9]. The German Dictionary published in 1809 interpreted the term “sustainability” as something that people can still rely on when everything else is unsustainable. It can be seen that the early thought of SD mainly reflected in the sustainable use of natural resources.”
The last paragraph in 3.3.4. United Nations Sustainable Development Summit needs to be deleted.
Response: Thank you so much for your careful checking. We have deleted the last paragraph in 3.3.4.
Headings, 4. Connotation interpretation of SD and 5. Perfecting process of the objectives of SD, may not be the best headings to describe what was written: 4 could be “Types (or levels) of SD”? 5 could be “Goals of SD”?
Response: Thank you very much for taking time to check our article and putting forward good advice. We think the headings you suggested are more proper for section 4 and 5 and have changed the headings according to your advice.
Section 4 appears to neglect social dimension of sustainability focusing on economy and environment.
Response: Thank you very much for taking time to check our article. We have mentioned the classification of natural capital, manufactured capital, human capital and social capital, and the social capital refers to the social dimension of sustainability. However, the 3 types of SD are mainly based on the substitution between natural and manufactured capital, which refer to economic and environmental dimensions respectively, so in the following we focused on economy and environment.
The second paragraph in 4.3. Absurdly strong sustainability could become separate sub-section, 4.4. Comparisons among weak, strong and absurdly strong sustainability. The discussion on advantages and disadvantages of weak, strong and absurdly strong sustainability should be more structured such that all dis/advantages of three types of sustainability are equally discussed.
Response: Thank you very much for your good advice. We have made the second paragraph in 4.3. a separate sub-section, that is 4.4 Comparisons among weak, strong and absurdly strong sustainability, and the discussion on advantages and disadvantages of weak, strong and absurdly strong sustainability have been added.
“4.4. Comparisons among weak, strong and absurdly strong sustainability
The advantage of weak sustainability is that it affirms the progress of science and technology, and the substitutability of natural and manufactured capital is more in line with the needs of economic development; The disadvantage of weak sustainability is that it is too optimistic about human's ability to control nature and technological progress, and believes nature has no constraint capacity and all the ecosystem functions can be replaced. The advantage of the absurdly strong sustainability is that the economic system is a subsystem of nature, rather than a system independent of nature; The disadvantage of the absurdly strong sustainability is that the role of technology is underestimated, and all the natural capital is thought to be restrictive, while in fact some natural capital can be replaced. For example, some mineral resources can be replaced by other man-made products. The advantage of strong sustainability is that it is against the above two extreme views, which is the concept of sustainability we should accept [10]; The disadvantage of strong sustainability is that it sets a series of thresholds that must not be crossed, which may hinder the economic development of especially developing countries.”
Please define Green GDP on page 7.
Response: We have added the definition of Green GDP on page 8 (original page 7), that is “which is calculated by subtracting the costs of environmental and ecological damage done in a specific period of time from GDP from that some time”.
In 5.1. Single-factor development goals, precursors of MDGs are missing such as a series of UN-led conferences in the 1990s, and other activities, documents and discussions by OECD, World Bank and IMF.
Response: Thanks very much for your advice. We have added more materials in 5.1.
“The term sustainable development came into prominence in 1980, when the International Union for the Conservation of Nature and Natural Resources (IUCN) presented the World Conservation Strategy (WCS) with "the overall aim of achieving sustainable development through the conservation of living resources"[11]. Then a series of influential actions, conferences, documents and discussions such as 21st century agenda paved the way for a global mindset on sustainable development [56]. Environmental issues are also recognized as the priority of supports and believed as the foundation and core of sustainable growth by international groups including IMF, World Bank [57,58]. Meanwhile, some SD assessment indicators were established including System of Integrated Environmental and Economic Accounting (SEEA), Ecological Footprint (EF). And Millennium Ecosystem Assessment (MA) from 2001-2005 assessed the consequences of ecosystem change for human well-being [12]. In short, SD in this period addressed mainly the issue of ecological sustainability with the goal of sustainable utilization of nature resources environmental protection [13].”
In 5.2. Millennium Development Goals (MDGs), most sentences appear to use wrong tenses: shouldn’t they be past or past perfect?
Response: Thanks for your careful checking. We have changed the tenses to past or past perfect.
In 5.3. Sustainable Development Goals (SDGs), some verb forms appear to be incorrect: please correct all the verbs according to the plural forms of MDGs and SDGs.
Response: Thanks for your careful checking. We have corrected all the verbs according to the plural forms of MDGs and SDGs in 5.3.
Some English expressions are unconventional: professional proofreading is required. For example, in the second paragraph of 5.2., “Armed conflicts, kidnappings, terrorist attacks, et al.” or in the second paragraph of 5.3., “Some comparations have been made”.
Response: Thanks for your careful checking. We have proofread the language.
Some sentences are overly long: please split these long sentences into two or more sentences. For instance, “Take China’s green GDP index as an example […] otherwise economic development is sustainable” or “Similarly, the “Genuine Saving” (GS) proposed […] but it still examined the total amount of capital” in 6.1.
Response: Thanks for your careful checking. We have split the long sentences.
In 6. Prospects of SD research, please expand how the authors selected the four issues and introduce the issues before 6.1.
Response: Thanks for your careful checking and good advice. We have added the reasons for selection before 6.1.
“According to the previous analysis, though strong sustainability is the concept people should accept, weak sustainability still remains. And although the goals of SD have developed from addressing ecological sustainability to more comprehensive goals, but they only consider SD in the next 15 years rather than a longer time. Furthermore, in 1992, the 3 pillars of SD were considered to be the economy, the society, and the environment. This has been expanded by some authors to include a fourth pillar of culture, institutions or governance [14–16]. Based on the above analysis, these issues were selected for further discussion.”
The expressions, “real concept of SD” or “real SD” should be reconsidered as the meaning is not clear: why real? Who defines what is real or not? Is it about ideal? Or sufficiently strong? Sustainable development?
Response: Thank you very much for taking time to check our article and putting forward good advice. We have changed the title according to your advice.
In section 4, there are 3 types of SD, and according to our analysis and the conclusions of other scholars, strong SD is the concept of SD we should accept, so we call strong SD “real concept of SD” or “real SD”. In order to make it clear, we have changed “real concept of SD” to “the concept of SD we should accept”, and “real SD” to “strong SD”.
Whole discussion and conclusions are missing. So what? What are the overall lessons learned? What do you suggest after your review?
Response: Thank you very much for taking time to check our article and putting forward good advice. We have added conclusions on page 12.
“7. Conclusion
The theory of SD appeared in 1980's, focusing on the coordinated development of economy, society and environment, and has entered the high-level political agenda. Now, SD theory has become an integral part of the agenda of governments and companies, and SD goals have become a core part of the mission of research institutions around the world [3]. The theory of SD has experienced different stages of development since it was put forward. At present, there are a lot of definitions of SD, but there are still misinterpretations. Based on the process of cognitive development, this paper combed the evolution of practice and theory, and concluded that the theory of SD has gone through 3 stages: the embryonic stage (before 1972), the molding stage (1972-1987) and the developing stage (1987 - present). The concept of SD has gradually evolved from the initial vague definition to a global action and has contained more and more practical wisdom. In the process of development, it is considered that strong SD which requires the increase of total capital and rationality of capital structure is the concept of SD people should accept. The goal of sustainable development has changed from pursuing the single factor goal of ecological sustainability to MDGs of focusing on poverty, until now it is a more comprehensive and universal SDGs. However, at present, there is still a confusion of strong and weak sustainability, inter-generational and intra- generational equity. Weak sustainability and the pursuit of short-term intra-generational equity still exist. In addition, considering local cultural factors, improving governance capacity and focus more on life support systems are considered as important means to promote sustainable development.”
In author contributions, FY’s contribution is missing: if this is not a mistake, s/he should be removed from the author list.
Response: We are sorry for missing FY’s contribution and we have corrected author contributions.
Reviewer 2 Report
The paper The Evolution Process, Problems and Research Prospects of Sustainable Development Theory consists of 6 sections:1 Introduction, 2 Status of research, 3 Evolution process of SD theory, 4 Connotation interpretation of SD , 5 Perfecting process of the objectives of SD, 6 Prospects of SD research.
Sustainable development is a very actual and important issues.However, I have three main remarks about this manuscript:
I cannot understand what is/are contribution/s of the paper?
The discussions about different topics are too simple; the investigation about the published articles is simple too. The evolution stages are also very simple and obvious.
The other main remark is about the “theory” and “practice” of SD.
In the introduction the authors stress the lack of understanding of SD, and various definitions and interpretations as a problem. – the paper does not bring more understanding in these topics.
Furthermore, the authors continue: “According to the principle of dialectical relationship between practice and cognition, and the principle of repetition and infinity of cognition process, SD practice is the foundation of SD theory. SD theory guides SD practice, and is constantly tested and perfected in practice (16)”. In the article (16) ‘research -based knowledge’ not ‘theory’ is the central term.
The third main remark is about topics in sections 4, 5 6: how they were identified? Way these topics, why not others?
There are already published articles about sustainable development research:
Kajikawa, Y., Ohno, J., Takeda, Y., Matsushima, K., & Komiyama, H. (2007). Creating an academic landscape of sustainability science: an analysis of the citation network. Sustainability Science, 2(2), 221.
Zhu, J., & Hua, W. (2017). Visualizing the knowledge domain of sustainable development research between 1987 and 2015: a bibliometric analysis. Scientometrics, 110(2), 893-914.
Wichaisri, S., & Sopadang, A. (2018). Trends and future directions in sustainable development. Sustainable Development, 26(1), 1-17.
Leal Filho, W., Azeiteiro, U., Alves, F., Pace, P., Mifsud, M., Brandli, L., ... & Disterheft, A. (2018). Reinvigorating the sustainable development research agenda: the role of the sustainable development goals (SDG). International Journal of Sustainable Development & World Ecology, 25(2), 131-142
Many other classic or relevant references are missing, such as for example
Robert, K. W., Parris, T. M., & Leiserowitz, A. A. (2005). What is sustainable development? Goals, indicators, values, and practice. Environment: science and policy for sustainable development, 47(3), 8-21. Citerat av 1765
Jacobus A. Du Pisani Professor of History (2006) Sustainable development – historical roots of the concept, Environmental Sciences, 3:2, 83-96, DOI: 10.1080/156934306006888
What about sustainability science? What about circular economy? What about climate change?
Sauvé, S., Bernard, S., & Sloan, P. (2016). Environmental sciences, sustainable development and circular economy: Alternative concepts for trans-disciplinary research. Environmental Development, 17, 48-56.
I have not checked other references, but the reference 21 is not correct:
Wu, J.; Guo, X.; Yang, Y.; Qia, G.; Niu, J.; Liang, C.; Zhang, Q.; Li, A. What is sustainability science? Chinese J. Appl. Ecol. 2014, 25, 1–11.
: Wu, J.; Guo, X.; Wang, Y.; Qiu, G.; Liu, J.; Liang, C.; Zhang, Q.; Li, A. What is sustainability science? Chinese J. Appl. Ecol. 2014, 25, 1–11.
Author Response
Reviewer 2
Comments:
The paper The Evolution Process, Problems and Research Prospects of Sustainable Development Theory consists of 6 sections:1 Introduction, 2 Status of research, 3 Evolution process of SD theory, 4 Connotation interpretation of SD, 5 Perfecting process of the objectives of SD, 6 Prospects of SD research.
Sustainable development is a very actual and important issues. However, I have three main remarks about this manuscript:
I cannot understand what is/are contribution/s of the paper?
The discussions about different topics are too simple; the investigation about the published articles is simple too. The evolution stages are also very simple and obvious.
Response: Thanks for your valuable comments. SD is an important topic and have been studied for years. Now, MDGs have expired and we are working for SDGs. In order to better guide the practice of SD, we think it is necessary to reorganize the evolution process of the practice and theory of SD. The purpose of this article is to review the evolution of SD theory including the developing stages, types and goals. Besides, we also work to bring the situation and concepts of China into the discussion. We have tried to collect as much as materials as we can and pointed out the concept people should accept. And we will go on working according to your advice.
The other main remark is about the “theory” and “practice” of SD.
In the introduction the authors stress the lack of understanding of SD, and various definitions and interpretations as a problem. – the paper does not bring more understanding in these topics.
Furthermore, the authors continue: “According to the principle of dialectical relationship between practice and cognition, and the principle of repetition and infinity of cognition process, SD practice is the foundation of SD theory. SD theory guides SD practice, and is constantly tested and perfected in practice (16)”. In the article (16) ‘research -based knowledge’ not ‘theory’ is the central term.
Response: Thank you for your valuable comments. The concept of SD emerged and became a basic strategy to guide the world's socio-economic transformation, however, the lack of understanding of SD is still a problem. We think the review of the evolution process of the practice and theory of SD may be useful for better understanding SD theory and guiding SD practice. Therefore, we summarized the types and goals of SD. We find that strong sustainability is the concept people should accept, and the goals of SD have developed from addressing only the issue of ecological sustainability to SDGs which are more comprehensive. In section 6, we selected some issues for further discussion. We wish these works could be helpful for better understanding of SD.
We confused “research-based knowledge” and “theory”, and we have changed the reference to [20] Niu, W. Theory and practice of China’s sustainable development. Bull. Chinese Acad. Sci. 2012, 27, 280–289.
The third main remark is about topics in sections 4, 5 6: how they were identified? Way these topics, why not others?
Response: Thanks for your taking time to check our article. The purpose of this article is to review the evolution of sustainable development theory. Distinguishing the types of SD is a basic issue to understanding the concept of SD. And the goals of SD are the reflection of the concept of SD. These are basic issues of SD and are also concerned by other scholars, so we selected these topics in sections 4,5. Moreover, we selected the issues in section 6 mainly based on the previous analysis on the evolution process of SD practice and theory and current research hotspots of SD. According to the previous analysis, though strong sustainability is the concept people should accept, weak sustainability still remains. And although the goals of SD have developed from addressing ecological sustainability to more comprehensive goals, but they only consider SD in the next 15 years rather than a longer time. Furthermore, since 1992, the 3 pillars of SD were considered to be the economy, the society, and the environment. Then culture or cooperative governance was considered to be the 4th pillar of SD. Based on the above analysis, these issues were selected for further discussion in section 6.
There are already published articles about sustainable development research:
Kajikawa, Y., Ohno, J., Takeda, Y., Matsushima, K., & Komiyama, H. (2007). Creating an academic landscape of sustainability science: an analysis of the citation network. Sustainability Science, 2(2), 221.
Zhu, J., & Hua, W. (2017). Visualizing the knowledge domain of sustainable development research between 1987 and 2015: a bibliometric analysis. Scientometrics, 110(2), 893-914.
Wichaisri, S., & Sopadang, A. (2018). Trends and future directions in sustainable development. Sustainable Development, 26(1), 1-17.(未引用)
Leal Filho, W., Azeiteiro, U., Alves, F., Pace, P., Mifsud, M., Brandli, L., ... & Disterheft, A. (2018). Reinvigorating the sustainable development research agenda: the role of the sustainable development goals (SDG). International Journal of Sustainable Development & World Ecology, 25(2), 131-142.
Response: Thanks for your kind sharing. We have read these papers carefully and think they are very helpful for our research. And we have added most of them in the reference.
There are already published articles about SD research. The first 3 listed articles mainly used bibliometric analysis. In the 1st article a topological clustering method was used to detect the subdomains of sustainability science; In the 2nd article, citespace was used to represent the knowledge structure and evolution of SD in the post-WCED era; The 3rd article aims to identify research clusters. However, these articles are not from the perspective of theoretical evolution. And the 4th article mainly focused on SDGs, aiming to look at the implementation of the SDGs and to delineate a set of research needs. We expect to make a more comprehensive review from types, goals and research prospects of SD. Besides, we also work to bring the situation and concepts of China into the discussion.
Many other classic or relevant references are missing, such as for example
Robert, K. W., Parris, T. M., & Leiserowitz, A. A. (2005). What is sustainable development? Goals, indicators, values, and practice. Environment: science and policy for sustainable development, 47(3), 8-21. Citerat av 1765
Jacobus A. Du Pisani Professor of History (2006) Sustainable development – historical roots of the concept, Environmental Sciences, 3:2, 83-96, DOI: 10.1080/156934306006888
Response: Thanks for your sharing. We are sorry for missing these classic references and have read these papers carefully. They are helpful for our research and we have added them in the reference.
What about sustainability science? What about circular economy? What about climate change?
Sauvé, S., Bernard, S., & Sloan, P. (2016). Environmental sciences, sustainable development and circular economy: Alternative concepts for trans-disciplinary research. Environmental Development, 17, 48-56.
Response: Thanks for your careful checking and putting forwards good advice. Sustainable development contains many aspects. We have added discussion about sustainability science in 3.3.3. From SD to sustainability science. these topics in the article. However, the paper focuses on the evolution of sustainable development theory including types, goals and research prospects, so there is no in-depth discussion on circular economy and climate change. We will keep on researching according to your advice in the future.
I have not checked other references, but the reference 21 is not correct:
Wu, J.; Guo, X.; Yang, Y.; Qia, G.; Niu, J.; Liang, C.; Zhang, Q.; Li, A. What is sustainability science? Chinese J. Appl. Ecol. 2014, 25, 1–11.
: Wu, J.; Guo, X.; Wang, Y.; Qiu, G.; Liu, J.; Liang, C.; Zhang, Q.; Li, A. What is sustainability science? Chinese J. Appl. Ecol. 2014, 25, 1–11.
Response: Thanks for your careful checking. The authors’ names were incorrect in reference 21 and we have corrected them.
Reviewer 3 Report
Thank you for your efforts to initiate a review of the most important global topic. Given the current status of the world's life support systems, I think now is a critical time for such a review. There is a great deal of good material in your draft article, particularly regarding the line of cognitive-political development around the SD concept. Critically, you also work to bring the situation and concepts of China into the discussion. That is incredibly important for the future of the planet. I suggest that the review needs to be much more holistic (整體). Regardless of the 'twist and turns' that have occurred since 1987 when the term was established, there can be no ignoring the fact that the concept, from its inception was and is inseparable from considering the status of global life support systems. The manuscript as it stands does not appropriately review the status and changes in that status since the inception of the SD term; without (for example only) any systematic review of where the world status of biodiversity depletion, climate change nor the overharvest of the oceans. The status of the world's life support systems - must - be taken into account in the review and projection of 'research prospects' within the SD concept - otherwise SD is purely an academic exercise that is more of a 'play on words' than a true connection to the planet that sustains all life as we know it. The review needs to also look more closely at the efficacy of SD over time - with respect to life support systems, biodiversity conservation and the destruction of ocean ecosystems globally; to name just a few of what should be a somewhat longer list of considerations. The statement that "By 2015, the 15 years’ time limit for MDGs has expired, and unprecedented results have been achieved worldwide" is surely based upon the advancement of some parameters but does not deal with planet earth in terms of SD. Perhaps it would be useful to include in your review, the consideration of where we are as a species in terms of balancing (平衡) the anthropocene and ecocentric aspects of our research and related programmatic agendas. Perhaps as a starting point on that, consider: DOI: 10.1111/1477-8947.12083 and the perspective that research and education are inextricably (千絲萬縷) linked.
The Angthropocene Era
Author Response
Reviewer 3
Comments:
Thank you for your efforts to initiate a review of the most important global topic. Given the current status of the world's life support systems, I think now is a critical time for such a review. There is a great deal of good material in your draft article, particularly regarding the line of cognitive-political development around the SD concept. Critically, you also work to bring the situation and concepts of China into the discussion. That is incredibly important for the future of the planet.
I suggest that the review needs to be much more holistic (整體). Regardless of the 'twist and turns' that have occurred since 1987 when the term was established, there can be no ignoring the fact that the concept, from its inception was and is inseparable from considering the status of global life support systems. The manuscript as it stands does not appropriately review the status and changes in that status since the inception of the SD term; without (for example only) any systematic review of where the world status of biodiversity depletion, climate change nor the overharvest of the oceans. The status of the world's life support systems - must - be taken into account in the review and projection of 'research prospects' within the SD concept - otherwise SD is purely an academic exercise that is more of a 'play on words' than a true connection to the planet that sustains all life as we know it. The review needs to also look more closely at the efficacy of SD over time - with respect to life support systems, biodiversity conservation and the destruction of ocean ecosystems globally; to name just a few of what should be a somewhat longer list of considerations. The statement that "By 2015, the 15 years’ time limit for MDGs has expired, and unprecedented results have been achieved worldwide" is surely based upon the advancement of some parameters but does not deal with planet earth in terms of SD. Perhaps it would be useful to include in your review, the consideration of where we are as a species in terms of balancing (平衡) the anthropocene and ecocentric aspects of our research and related programmatic agendas. Perhaps as a starting point on that, consider: DOI: 10.1111/1477-8947.12083 and the perspective that research and education are inextricably (千絲萬縷) linked.
Response:
Thank you so much for your valuable comments and we have revised the paper according to your advice. Details are as follows:
We have added the status of world's life support systems in the 1. Introduction “Meanwhile, preserving the global life support systems is made more difficult by the rapid and continuing global environmental changes in the air, oceans, land, and freshwater systems due to human actions [8].”and 6.5 Global life supporting system in 6 Prospects of SD research.
What’s more, human beings as a species in terms of balancing the anthropocene and ecocentric aspects of SD is an important topic. We have discussed this issue in 6.5 Global life supporting system. The article DOI: 10.1111/1477-8947.12083 has adopted “Yin and Yang” philosophy from ancient China to seeking balance between these ecocentric and anthropocentric paradigms. But limited to the length of the paper, there is no in-depth discussion. Thanks very much for your good advice and we will do more research about it in the future.
“6.5. Global life supporting system
Though the focus on earth's life supporting system cannot replace efforts to eradicate poverty, improve health and other ways to improve the quality of life of vulnerable groups, the stable functioning of global life supporting system is the prerequisite for human development [8,90]. However, since 2000, accumulating research shows that this functioning is at risk, and that further human pressure may lead to changes in climate, biodiversity, land use, et al [91,92]. Some research shows that more than 50% of Earth’s land surface has been directly modified by human action by 2012 [93]. MDGs concentrated largely on social outcomes; Reaching beyond MDGs, SDGs explicitly incorporate objectives to preserve natural ecosystems [90]. To achieve this goal, finding ways such as Chinese Yin-Yang approach to balance the anthropocene and ecocentric aspects of SD is an important issue [94].”
Round 2
Reviewer 1 Report
Please proofread the manuscript to improve general English, fonts, etc.
Author Response
Thank you for taking the time to check our article. We have polished up the language of the article and checked the fonts and format.
Reviewer 2 Report
There is so many publications about SD. In my opinion this manuscript does not offer any new insights. It goes not in the depth in any topic either. The authors made some changes, but the scope of the manuscript is still the same.
I still cannot understand what is/are contribution/s of the paper. The discussions about different topics are too simple; the investigation about the published articles is simple too. The evolution stages are also very simple and obvious.
Author Response
Reviewer 2
Comments:
There is so many publications about SD. In my opinion, this manuscript does not offer any new insights. It goes not in the depth in any topic either. The authors made some changes, but the scope of the manuscript is still the same.
I still cannot understand what is/are contribution/s of the paper. The discussions about different topics are too simple; the investigation about the published articles is simple too. The evolution stages are also very simple and obvious.
Response: Specials thanks to you for taking time to check our article and your valuable comments.
There are already published many articles about SD research, also as shown in this paper. However, these articles mainly focus on the SDGs or the environmental or economical technologies to achieving SD. For realizing the SD practices and UN SDGs, the study of SD theory has very important guiding significance. The articles from the perspective of theoretical evolution are relatively scarce. Meanwhile, according to “2. Status of SD research”, there are relatively few studies on SD theory between 2000-2019. Therefore, it is necessary to reorganize the evolution process of the theory and practice of SD.
Based on the analysis of the gradual evolution and the recent improvement process of the concept and objective of SD, the main purpose of the article is to put forward some prospects that can promote the study of SD. The results show that SD is gradually implemented into a global action from the initial fuzzy concept, including increasing practical wisdom; the goal of SD evolves from pursuing the single goal of sustainable use of natural resources to Millennium Development Goals (MDGs) and Sustainable Development Goals (SDGs). This paper argues that the theory of strong sustainability should be the accepted concept of SD. And culture, good governance and life support systems are important factors to promote SD.
We have added some in-depth discussion and summary in “4.4 Comparisons among weak, strong and absurdly strong sustainability” and “6.1. Weak sustainability remains” to strength the understanding of SD. Limited by the article space, it’s difficult to go in depth in every topic. We will go on researching one certain topic in depth in future.
Once again, thank you very much for your comments and suggestions.
Reviewer 3 Report
The review is shaping up quite nicely. The additions that you provided in the newest version support the paper, but may need to be better organized into the flow of the manuscript. The use of Chinese translation greatly supports the foundations of the study in China while analyzing/reflecting upon global developments. This could perhaps be expanded upon a bit further. Further support on the use of English within the context of the author's perspectives could be useful for international readability.
Author Response
Reviewer 3
Comments:
The review is shaping up quite nicely. The additions that you provided in the newest version support the paper but may need to be better organized into the flow of the manuscript. The use of Chinese translation greatly supports the foundations of the study in China while analyzing/reflecting upon global developments. This could perhaps be expanded upon a bit further. Further support on the use of English within the context of the author's perspectives could be useful for international readability.
Response: Special thanks to you for your good comments.
We mainly added several sections in the manuscript. (1) The method section. We added the method of the whole reaearch in the last paragraph in “1. Introduction” and the method of the literature retrieval in the first paragraph in “2. Status of SD research”. (2) “3.3.3. From SD to sustainability science”. We put it in section 3 to perfect the evolution process of SD theory. (3) The reasons why we selected these issues for discussion. We put in in the beginning of “6. Prospects of SD research”. (4) “6.5. Global life-supporting system” We put it in section 6 as one of the perspects of future research.
And we have reorganized the addtion about life supporting systems in “1. Introduction” as follows: “Since the start of the Industrial Revolution, the population has increased rapidly and production has been developing. Human beings have been exploiting wealth from nature and the volume of wastes and pollutants thrown into the environment has also gradually been increased. Preserving the global life support systems has become more difficult due to the rapid and continuing human-caused environmental changes [8].”
Though the terms “sustainability” or “sustainable” was not put forward by Chinese, the idea of SD has a long history in China. Therefore, we used the examples in China to support and supplement the views in the article. However, the article focuses on the evolution of SD theory worldwide. Therefore we also added examples in the ancient Egyptian, Mesopotamian, Greek and Roman civilizations, as well as in German to better reflect the global development of SD in the embryonic period. We have polished up the language of the article and checked the fonts and format.
Once again, thank you very much for your comments and suggestions.
Round 3
Reviewer 2 Report
Dear authors and Editors, the manuscript has been improved, but not significantly.
1)Frist of all, a publication on trends in research on sustainable development in a journal with focus on sustainability should be excellent, not pure.
2)There are already published articles with review of sustainable development research. The literature review of these publications in the manuscript is not made.
The authors state “2. Status of SD research”, there are relatively few studies on SD theory between 2000-2019. Therefore, it is necessary to reorganize the evolution process of the theory and practice of SD.”
Belowe examples on publications:
Kajikawa, Y., Ohno, J., Takeda, Y., Matsushima, K., & Komiyama, H. (2007). Creating an academic landscape of sustainability science: an analysis of the citation network. Sustainability Science, 2(2), 221.
Zhu, J., & Hua, W. (2017). Visualizing the knowledge domain of sustainable development research between 1987 and 2015: a bibliometric analysis. Scientometrics, 110(2), 893-914.
Wichaisri, S., & Sopadang, A. (2018). Trends and future directions in sustainable development. Sustainable Development, 26(1), 1-17.
Leal Filho, W., Azeiteiro, U., Alves, F., Pace, P., Mifsud, M., Brandli, L., ... & Disterheft, A. (2018). Reinvigorating the sustainable development research agenda: the role of the sustainable development goals (SDG). International Journal of Sustainable Development & World Ecology, 25(2), 131-142
I still can not see contribution of this paper. The authors state:
"This paper argues that the theory of strong sustainability should be the accepted concept of SD. And culture, good governance and life support systems are important factors to promote SD. "
My question is: how this is related to empirical results in this manuscript???
Furthermore: Arguing for strong sustainability requires also reviewing of relevant literature. See for example: Liobikiene, G., Balezentis, T., Streimkiene, D., & Chen, X. (2019). Evaluation of bioeconomy in the context of strong sustainability. Sustainable Development.
Other examples regard decoupling and economic growth: Fletcher, R., & Rammelt, C. (2017). Decoupling: A key fantasy of the post-2015 sustainable development agenda. Globalizations, 14(3), 450-467; Ward, J. D., Sutton, P. C., Werner, A. D., Costanza, R., Mohr, S. H., & Simmons, C. T. (2016). Is decoupling GDP growth from environmental impact possible?. PloS one, 11(10);
Barbier, E. B. (2016). Is green growth relevant for poor economies?. Resource and energy economics, 45, 178-191; Jason Hickel (2019) Is it possible to achieve a good life for all within planetary boundaries?, Third World Quarterly, 40:1, 18-35, DOI: 10.1080/01436597.2018.1535895
3) Finally, the quality of how different issues/ topics which treated in the manuscript is really very pure. There are already a lot of different other publications on sustainable development.
Just only one example: “Sustainable development: nuances and perspectives”
Author Response
Reviewer 2
Comments:
Dear authors and Editors, the manuscript has been improved, but not significantly.
Frist of all, a publication on trends in research on sustainable development in a journal with focus on sustainability should be excellent, not pure.
Response:Thanks for your valuable comments.
The main purpose of this article is to better understand SD theory by reviewing the evolution of SD theory including the developing stages, types and goals. Besides, we also worked to bring the situation and concepts of China into the discussion. We have tried to collect as much as materials as we can and put forward the concept people should accept as well as 6 issues which we think need further discussion. We wish these works could be helpful for better understanding of SD theory and guiding SD practice.
There are already published articles with review of sustainable development research. The literature review of these publications in the manuscript is not made.
(1)The authors state “2. Status of SD research”, there are relatively few studies on SD theory between 2000-2019. Therefore, it is necessary to reorganize the evolution process of the theory and practice of SD.”
Belowe examples on publications:
Kajikawa, Y., Ohno, J., Takeda, Y., Matsushima, K., & Komiyama, H. (2007). Creating an academic landscape of sustainability science: an analysis of the citation network. Sustainability Science, 2(2), 221.
Zhu, J., & Hua, W. (2017). Visualizing the knowledge domain of sustainable development research between 1987 and 2015: a bibliometric analysis. Scientometrics, 110(2), 893-914.
Wichaisri, S., & Sopadang, A. (2018). Trends and future directions in sustainable development. Sustainable Development, 26(1), 1-17.
Leal Filho, W., Azeiteiro, U., Alves, F., Pace, P., Mifsud, M., Brandli, L., ... & Disterheft, A. (2018). Reinvigorating the sustainable development research agenda: the role of the sustainable development goals (SDG). International Journal of Sustainable Development & World Ecology, 25(2), 131-142.
Response: Thanks for your helpful and valuable comments.
We have studied these articles carefully. The first 3 articles mainly used bibliometric analysis to review the study of SD, and the 4th article focused on SDGs, aiming to look at the implementation of the SDGs and to delineate a set of research needs. These are valuable references, and we have cited them in section 2, section 3.3.3 and section 5.3. Except for bibliometric analysis, we also make a more comprehensive review from types, goals and research prospects of SD and bring the situation and concepts of China into the discussion.
Besides, the sentence “There are relatively few studies on SD theory or the combination of theory and practice” in section 2 is not rigorous. We changed it to “The studies on SD theory are fewer than SD practice”. And the purpose of the article is to re-carding the evolution of SD theory and put forward the research focuses in the future.
Thanks again for your kind sharing.
(2) I still can not see contribution of this paper. The authors state:
"This paper argues that the theory of strong sustainability should be the accepted concept of SD. And culture, good governance and life support systems are important factors to promote SD. "
My question is: how this is related to empirical results in this manuscript???
Response:Thanks for your taking time to check our article.
This paper argues that the theory of strong sustainability should be the accepted concept of SD, and culture, good governance and life support systems are important factors to promote SD. These conclusions are related to the previous analysis on the evolution process of SD practice and theory and current research hotspots of SD.
First, section 4 discussed different types of SD. Through the comparisons among them, we found that the views of weak sustainability and absurdly strong sustainability are too extreme. Strong sustainability is opposed the above two extreme views and is the concept people should accept., which we also argued in section 7. Conclisuon.
Second, in section 3.3, since 1992, the 3 pillars of SD were the economy, the society, and the environment. By the end of the 2012 United Nations Conference, SD expanded from three pillars to four: economic, social, environmental and governance. Moreover, culture or cooperative governance was also considered to be the 4th pillar of SD. So, the paper argued that culture and good governance were important factors to promote SD.
Third, in section 1 Introduction, the paper said that preserving the global life support systems has become more difficult due to the rapid and continuing human-caused environmental changes. In section 5 Goals of SD, through the comparison of SDGs and MDGs, we found that MDGs concentrated largely on social outcomes, while SDGs called on people not to survive at the cost of damaging the ecological environment, which explicitly incorporated objectives to preserve natural ecosystems. In section 6.5, we argued that the stable functioning of the global life-supporting system is a prerequisite for human development. MDGs concentrated largely on social outcomes; unlike MDGs, SDGs explicitly incorporate objectives to preserve natural ecosystems. So, the paper argued that life support systems are important factors to promote SD.
(3)Furthermore: Arguing for strong sustainability requires also reviewing of relevant literature. See for example: Liobikiene, G., Balezentis, T., Streimkiene, D., & Chen, X. (2019). Evaluation of bioeconomy in the context of strong sustainability. Sustainable Development.
Other examples regard decoupling and economic growth: Fletcher, R., & Rammelt, C. (2017). Decoupling: A key fantasy of the post-2015 sustainable development agenda. Globalizations, 14(3), 450-467; Ward, J. D., Sutton, P. C., Werner, A. D., Costanza, R., Mohr, S. H., & Simmons, C. T. (2016). Is decoupling GDP growth from environmental impact possible?. PloS one, 11(10);
Barbier, E. B. (2016). Is green growth relevant for poor economies?. Resource and energy economics, 45, 178-191; Jason Hickel (2019) Is it possible to achieve a good life for all within planetary boundaries?. Third World Quarterly, 40:1, 18-35, DOI: 10.1080/01436597.2018.1535895
Response:Thanks for your helpful and valuable comments.
The article “Liobikiene, G.; Balezentis, T.; Streimikiene, D.; Chen, X. Evaluation of bioeconomy in the context of strong sustainability. Sustain. Dev. 2019, 27, 955–964.” is a valuable reference about strong sustainability and we have cited it in section 4.2 to supplement the discussion on strong sustainability that it “additionally requires not crossing ecological thresholds”.
The article “Barbier, E.B. Is green growth relevant for poor economies? Resour. Energy Econ. 2016, 45, 178–191.” discussed how the developing countries could achieve green growth. It provides good ideas for developing countries to achieve SD. The other 3 articles about decoupling and economic growth argue that it is impossible to decouple GDP growth from environmental impact. These are excellent materials to prove the rationality of strong sustainable development and provide strategies of SD for developed and developing countries. And we have cited them in section 6.1.
Thanks again for your kind sharing.
Finally, the quality of how different issues/ topics which treated in the manuscript is really very pure. There are already a lot of different other publications on sustainable development.
Just only one example: “Sustainable development: nuances and perspectives”
Response:Thanks for your valuable comments.
The book “Sustainable development: nuances and perspectives” focus on the distinctions among the different understandings of SD and offers an appreciation of the diversity, and is helpful for revising and improving my manuscript. We have cited it in the 2nd paragraph in section 1 prove the importance of reviewing the evolution of SD theory.
Besides the distinctions among the different understandings of SD, we also focused on the evolution of SD theory and the concept and the way of development we should accept. As a review, our article put forward some prospects that can promote the study of SD based on the analysis of the gradual evolution and the recent improvement process of the concept and objective of SD. Limited by the article space, it’s difficult to go in depth in every topic.
Thanks again for your kind sharing.